# The Role of Microglia in Neuroinflammation of the Spinal Cord after Peripheral Nerve Injury

**DOI:** 10.3390/cells11132083

**Published:** 2022-06-30

**Authors:** Tana S. Pottorf, Travis M. Rotterman, William M. McCallum, Zoë A. Haley-Johnson, Francisco J. Alvarez

**Affiliations:** 1Department of Cell Biology, Emory University, Atlanta, GA 30322, USA; tana.pottorf@emory.edu (T.S.P.); will8mccallum@emory.edu (W.M.M.); zoe.haley-johnson@emory.edu (Z.A.H.-J.); 2School of Biological Sciences, Georgia Institute of Technology, Atlanta, GA 30318, USA; travis.rotterman@biosci.gatech.edu

**Keywords:** microglia, axotomy, sensory neurons, motoneurons, neuroinflammation, neuroprotection, regeneration, synaptic plasticity

## Abstract

Peripheral nerve injuries induce a pronounced immune reaction within the spinal cord, largely governed by microglia activation in both the dorsal and ventral horns. The mechanisms of activation and response of microglia are diverse depending on the location within the spinal cord, type, severity, and proximity of injury, as well as the age and species of the organism. Thanks to recent advancements in neuro-immune research techniques, such as single-cell transcriptomics, novel genetic mouse models, and live imaging, a vast amount of literature has come to light regarding the mechanisms of microglial activation and alluding to the function of microgliosis around injured motoneurons and sensory afferents. Herein, we provide a comparative analysis of the dorsal and ventral horns in relation to mechanisms of microglia activation (CSF1, DAP12, CCR2, Fractalkine signaling, Toll-like receptors, and purinergic signaling), and functionality in neuroprotection, degeneration, regeneration, synaptic plasticity, and spinal circuit reorganization following peripheral nerve injury. This review aims to shed new light on unsettled controversies regarding the diversity of spinal microglial-neuronal interactions following injury.

## 1. Introduction

Peripheral nerve injury (PNI) results in an extensive immune reaction within the spinal cord. Ipsilateral to the injury, microglia, the native macrophage of the central nervous system (CNS), proliferate and undergo phenotypic changes that result in the release of pro- and anti-inflammatory cytokines and a variety of interactions with different spinal cellular elements. This immune reaction displays different intensities at specific locations in the dorsal and ventral horns. In the superficial lamina of the dorsal horn (lamina I to III (LI–LIII)), where the central projections of injured nociceptor afferents are localized, the immune reaction is very strong [1,2]. The microglia reaction is less extensive in deeper dorsal horn laminae. Finally, the ventral horn contains the cell bodies and proximal dendrites of axotomized motoneurons (MNs), as well as the central projections of peripherally injured proprioceptive sensory afferents [3,4,5]. Overall, most microglia throughout the ipsilateral ventral horn show morphologies suggesting activation, but their density and markers of activation are greatest around injured motoneurons in LIX. Injured proprioceptive afferents also travel in the medial deep dorsal horn (LV), where there is also a significant microglia reaction.

The dorsal horn microglial reaction has been thoroughly studied as it relates to neuropathic pain [6,7,8,9,10]. The ventral microglial reaction has also been well described regarding its properties and variations; however, its purpose and functional significance are less clear. In this review, we will compare microglia activity in the ventral and dorsal horns, focusing on microglial interactions with ventral MNs and the central branches of sensory afferents axotomized after PNI. We will compare mechanisms of microglial activation and examine how this reaction might impair or facilitate the recovery of motor function following the regenerative process. To best discuss microglia-MN interactions we will also draw from the extensive literature on brainstem motor facial and hypoglossal nuclei.

### 1.1. Peripheral Nerve Injuries

Peripheral nerves are the “conducting units” between the CNS and the rest of the body; they carry both afferent sensory information into the spinal cord and efferent motor commands to the periphery [11]. PNIs are common, affecting an estimated 1.6–2.8% of the population [12,13,14,15]. PNIs can occur due to physical trauma [16], chemical injury, or disease [17] (e.g., peripheral neuropathies that may be genetic or secondary to other diseases such as diabetes, ischemia, or autoimmune conditions). It is important to note that the distribution, properties, and timing of the microglial reaction depend on the type of nerve injury and its severity. We focus this review on animal models of physical nerve injury. Seddon (1943) was the first to classify traumatic nerve injuries into three categories: neuropraxia, axonotmesis, and neurotmesis [18]. Neuropraxia is the least severe and consists of temporary demyelination and blockade of impulse transmission, but no axotomy or Wallerian degeneration [18]. Axonotmesis is an injury that results in severed axons, damaged myelin sheaths, and Wallerian degeneration of the distal axon segments, but maintains intact perineurium and/or epineurium allowing for high efficiency and specificity of target reinnervation during nerve regeneration and high rates of functional recovery (i.e., crush injury) [18]. Neurotmesis is the most damaging and generally involves full transection of the nerve, cutting axons, myelin sheaths, and the nerve protective layers (endoneurium, perineurium, and epineurium), thus disconnecting the different nerve fascicles and frequently creating gaps between the distal and proximal stumps of the cut nerve. Neurotmesis requires nerve repair surgery and despite great advances in microsurgical techniques and bridging injury gaps, reinnervation of target tissues is frequently slow and inefficient [15,18,19,20,21,22]. Further differences arise because of the type of nerve and injury location (large or small, proximal, or distal, sensory, or motor, spared nerves, or complete injuries). We will review how nerve injury severity and its prognosis, in terms of potential for efficient functional nerve regeneration, influences the properties of microgliosis and neuroinflammation inside the spinal cord. The extension of the microglia reaction distributes to locations with axotomized MNs or injured sensory afferents. Thus, injury to the tibial nerve or sciatic nerve causes a larger microglia reaction than when the injury is restricted to the medial gastrocnemius nerve which contains fewer motor and sensory axons [3,23]. Similarly, a sciatic nerve injury will affect less territory in the Lumbar 5 (L5) segment dorsal horn than an L5 spinal nerve injury, but it will have a larger rostro-caudal distribution. Somatotopic and musculotopic relations of the injured nerves define the distribution of activated microglia in the spinal cord medio-lateral and rostro-caudal axes.

Following PNI, sensory and motor axons regenerate and reinnervate peripheral targets. This makes them distinct from axons injured in the CNS in which glial scarring and inhibiting molecules typically prevent axon regeneration [24]. The peripheral nerve promotes axon growth and regeneration, as can be demonstrated by bridging CNS lesions with peripheral nerve grafts [25,26,27]. However, many factors influence the ability of peripheral axons to regenerate and successfully reinnervate targets, including the type and location of the injury, the age of the individual, and the severity of the injury [28]. Even after regeneration, many patients commonly suffer persistent motor and sensory deficits. Slow speed of axon growth and errors in axon targeting in the periphery are important [29,30,31,32], but we will focus here on changes occurring centrally inside the spinal cord that are influenced by the central neuroinflammatory reaction orchestrated by microglia.

Central plasticity influences three different categories of mechanisms: neuroprotection, promoting axon regeneration, and central synaptic plasticity. PNIs cause varying degrees of neuronal loss (both motor and sensory) depending on the type and proximity of the injury [33,34,35]. Neuronal cell death causes an irreversible loss of circuitry that impedes complete functional recovery. In addition, spinal circuits are not “hardwired” and can undergo significant plasticity following PNI, resulting in abnormal sensorimotor processing, even in the absence of cell death. One example is the hyperexcitability of the dorsal horn, where nociceptive information is processed, resulting in hyperalgesia and chronic pain [36]. Another is the withdrawal of proprioceptive synaptic inputs from the ventral horn causing a permanent loss of the stretch reflex and altering the operation of critical spinal motor circuits [23,37,38,39,40,41]. Circuit/synaptic plasticity around injured motoneurons has historically been attributed to the neuroinflammatory response, but recent studies suggest greater nuance and complexities of the microglia reaction around axotomized MNs and sensory afferents injured in the periphery [42].

### 1.2. Microgliosis around Axotomized Motoneurons

In 1968, Blinzinger and Kreutzberg reported the presence of reactive microglia around axotomized MNs after injury to the facial nerve (a pure motor nerve). They performed a complete nerve transection and used electron microscopy (EM) to investigate changes in synaptic density on the surface of axotomized MNs four days after injury. They found an 80% reduction of synaptic boutons on the MN cell body and proximal dendritic arbor. This loss was concurrent with the proliferation and attraction of microglia to the surface of the MN cell body and very proximal dendrites [43]. Microglia processes were interwoven between synaptic boutons and the cell body of MNs. These spatial relations led to the hypothesis that microglial processes displace synaptic boutons and sever connectivity. However, Blinzinger and Kreutzberg never identified engulfment or phagocytosis of displaced boutons. They also suggested an alternative hypothesis: that metabolic changes and MN cell-autonomous processes could lead to the weakening and displacement of synaptic boutons, rather than their active removal by microglia. In this case, synapses would withdraw first, followed by microglia migration to occupy the available space. This report led to decades of debate about the functional roles of the microglia reaction around injured MNs. Similar microglia enwrapping was thereafter reported for every type of cranial and spinal MN investigated after axotomy [42]. Its significance related to survival, regeneration, and/or degeneration in addition to synaptotoxic or synaptoprotective actions have all been frequently debated [5]. A recent electron microscopy study combined with analyses of various markers of degeneration, complement, exosomes, and lytic or necroptotic vesicles provided compelling evidence that in a mouse model of unilateral sciatic nerve transection, microglia surrounding injured spinal MNs are actively removing disrupted and degenerated remains of synapses around the cell bodies of axotomized MNs, but without direct engulfment of intact synapses [44]. The significance of microglia seeking and attaching to the cell bodies of injured MNs is far from settled.

### 1.3. Microgliosis in the Dorsal Horn around the Central Projections of Sensory Afferents Axotomized in the Periphery

The proliferation of microglia in the spinal cord dorsal horn after PNI was first documented in the late 1970s [45,46] though it was decades later when this immune response was correlated with the onset and maintenance of pain states [6,7,8,9,10]. Since then, dorsal horn microglia have been the focus of extensive studies because of their contribution to neuropathic pain. Thus, microglial activation in the dorsal horn is frequently studied in the context of specific nerve injuries that reliably trigger hyperalgesia, allodynia, and augment dorsal horn excitability [47,48,49,50]. These include L5 spinal nerve injury/ligations, injuries to sciatic branches (tibial/peroneal) sparing the sural nerve, or chronic constriction injuries (CCI) of large limb nerves. It is not the goal of this review to summarize this vast literature but to compare, when relevant, differences or similarities between ventral horn microglia mechanisms with the much larger literature on dorsal horn microglia.

It is of interest that most nerve manipulations inducing dorsal horn microgliosis and pain also generate microgliosis in the ventral horn, even though in some experimental injuries there is no axotomy of motor axons. A recent study inducing neuropathic pain by CCI of the sciatic nerve reported significant differences in microglial activation between dorsal and ventral horn microglia [51]. Remarkably, in this injury model, MNs are not axotomized but still become surrounded by microglia, implying that this microglia behavior is not specific to axotomy and/or axon regeneration. Compared to dorsal horn microglia, microglia around MNs expressed less Brain-Derived Neurotrophic Factor (BDNF), fewer phagocytic markers and elements of complement cascades, but significantly more interleukin-6 (IL-6). There were also differences in morphological activation. Microglia extensively proliferate in both dorsal and ventral horns, but after CCI ventral horn microglia become more hypertrophied than dorsal horn microglia. Moreover, microglia tropism for neuronal cell body surfaces is not as conspicuous in dorsal horn microglia as in microglia surrounding injured MNs. Some dorsal horn microglia associate with myelinated axons that presumably originate in afferents injured in the nerve. One remarkable characteristic of the reaction of dorsal horn microglia to PNI is the differences between the sexes. Sex differences in pain biology have been abundantly reported, and recent work suggests the immune system, including, microglia, is a significant contributor to these differences [52,53,54,55]. Microgliosis is equally present in the dorsal horn after PNI in both males and females, however, the contributions of activated microglia to ongoing pain behaviors are more prevalent in males [52,56,57,58]. Sex differences have yet to be reported in the context of the ventral horn microglia reaction to PNI, their actions on motor circuits, or microgliosis around injured MNs.

### 1.4. Scope of the Present Review

Just over ten years ago, contemporary knowledge about the central microglia reaction after PNI was summarized in a comprehensive review by Professor Hakan Aldskogius [5]. The present review follows in the heels of this work focusing on new data that hopefully shed new light on mechanisms and past controversies on the microglia activation in the ventral horn, and its significance. We call attention to the diversity of activation states that have been described and the difficulty of considering spinal activated microglia in a monolithic fashion. Likely, the major change in the microglia field over the last ten years has been the proliferation of single-cell transcriptomic studies highlighting microglia heterogeneity, both in resting/surveying and activation states [59]. Not surprisingly a large heterogeneity of microglia activation profiles is being discovered within the spinal cord after nerve injuries. A second major change has been the availability of genetic mouse models to label microglia with fluorescent proteins that in combination with two-photon microscopy allow direct observation of dynamics and behavior [60]. Finally, there has been a bewildering proliferation of mouse models for microglia-specific manipulation. Here we summarize the contributions of these recent advancements to the outstanding controversies in the field of neuroinflammation following PNI.

## 2. Mechanisms of Spinal Microglial Activation, Motility, and Recruitment

### 2.1. Microglia Activation Dynamics in the Spinal Cord after Peripheral Nerve Injury

Resident microglia of the spinal cord are like microglia elsewhere in the CNS in that they are small cells with fine ramified processes that continuously scan areas of around 50–75 µm in diameter in well-defined territories separated from each other. Microglia processes extend and retract at a velocity of two to three micrometers per minute and transiently (for a few minutes) touch neuronal cell bodies, dendrites, axons, and synapses; these transient contacts are interpreted as monitoring homeostasis and activity in specific neuronal structures [61,62,63,64,65]. The CNS is surveyed in its entirety by microglial processes about three to four times per day [65]. Microglia cell bodies remain static at the center of their unique territories with their regular distribution maintained by cell-cell repulsion mechanisms between their processes [66]. After PNI, microglia proliferate extensively the first few days after injury [67,68]. Repulsion mechanisms disappear leading to an increase in microglia density. Activated and new microglia exhibit shorter, thicker, and more ramified processes that now overlap extensively with each other. PNI-activated spinal microglia do not transform, however, into process-lacking, ameboid, intensely phagocytic cells that are typical after spinal cord injury or other direct CNS traumatic injuries. To be sure, phagocytic markers such as CD68 increase expression after PNI [69], but they are variably regulated depending on the specific neural elements microglia interact with. Some microglia migrate towards the cell bodies of axotomized MNs or the central myelinated axons of sensory afferents injured peripherally, while others remain detached from these injured elements. The density and polarization of CD68 granules within microglia depend on the type of cell-cell interaction, as reviewed below. PNI-activated microglia also upregulate and release various cytokines and trophic factors, and these also show differences depending on their local environment in the dorsal and ventral horn. Finally, there is significant variation depending on the type, proximity, severity, and time after nerve injury. In other words, microglia-activated phenotypes are not globally switched on and off in binary fashion by PNI, but instead are graded, evolving, and modulated by signals in the local environment.

Some activated microglia in the ventral horn move towards the cell bodies of axotomized MNs and enwrap them. Two-photon microscopy revealed a two-step process [69]; during the first week, microglia extended processes towards the MN surface that end in phagocytic cups and then retract. During the second week, some microglia attach their cell body and primary processes to the MN surface, concentrating phagocytic markers (CD68) at the contact surface with the MN, but without damaging it. In addition, they develop thinner and shorter processes that resemble filopodia and scan the MN surface. Yet, other activated ventral horn microglia remain separated from the MN, with some interacting with the central axons of proprioceptors damaged in the peripheral nerve. These microglia do not polarize CD68 granules to specific regions of their membrane surface. The microglia reaction in the dorsal horn differs in that they do not enwrap neuronal cell bodies. However, some can partially surround myelinated axons, presumably from sensory neurons damaged in the periphery [70]. In the mouse spinal cord, it is not uncommon to find degenerating MNs after PNI. These MNs are completely surrounded by microglia forming large cell clusters (or nodules). Microglia in these clusters lack long processes and anastomose to each other, attracting infiltrating T-cells [69]. Similar microglia clusters were first described in the facial nucleus after facial nerve injuries that trigger MN cell death more readily than in the spinal cord [71]. Their frequency increases in situations of augmented MN cell death (root avulsion, injuries in SOD1 mutants, etc.) [72]. As reviewed below, microglia in these clusters are highly phagocytic and display unique properties.

Figure 1 summarizes the many microglia activation profiles found in the spinal cord after PNI and their relations to different spinal cord cellular elements. Spatially restricted cell-to-cell signals likely modulate differences according to location, specific neuronal element interactions, or time after injury. The following sections review signals that have been proposed to induce microglia proliferation, chemotaxis, phagocytosis, and expression of various cytokines and trophic factors after different nerve injuries.

### 2.2. Colony Stimulating Factor 1 (CSF1) and Other DNAX Activating Protein of 12kDa (DAP12) Activating Pathways

Recent evidence has shown that colony-stimulating factor 1 (CSF1; also known as macrophage colony-stimulating factor MCSF) is a critical signal responsible for the activation of microglia inside the spinal cord after PNI [3,57,73]. CSF1 is a cytokine that exclusively binds to the CSF1 receptor (CSF1-R). Within the CNS, the CSF1-R is expressed primarily by microglia [74]. After CCI of the sciatic nerve, both dorsal root ganglia sensory neurons and MNs upregulate CSF1. Specific deletion of CSF1 from sensory neurons, using an Advillin-Cre driver crossed with a floxed *Csf1* mouse (*Csf1^fl/^**^fl^*), reduced microglia proliferation in the dorsal horn after CCI, while not altering the ventral horn microglia response [57]. Eliminating all possible neuronal and glial CSF1 using a Nestin-Cre driver resulted in a decrease in the ventral horn microglia response as well [57]. Reduced dorsal horn microgliosis was associated with less severe nerve injury-induced mechanical hypersensitivity after CCI. We further supported the local role of CSF1 by confirming that MNs are the only spinal cells that increase expression of CSF1 after sciatic nerve transection and specific deletion of CSF1 from MNs using a choline acetyltransferase (ChAT)-Cre driver abrogated the ventral microglia response around axotomized MNs, not affecting dorsal horn microgliosis [3,73]. Microglia morphological changes and migration towards the cell bodies of injured MNs were all abolished. Thus, CSF1 is necessary for microglial activation after different forms of PNI, but the source of CSF1 is local and differs in the dorsal and ventral horns, originating, respectively, from injured sensory neurons or MNs.

Loss-of-function studies demonstrate that CSF1 from injured neurons is necessary for activating spinal microglia after PNI. They also explain an earlier study that showed a lack of microglia proliferation around axotomized MNs after facial nerve injury in osteopetrotic *op/op* mice carrying a spontaneous inactivating mutation in the *Csf1* gene [71]. The necessity of CSF1-R for the reaction of microglia around spinal MNs was recently confirmed by oral administration of the CSF1-R inhibitor PLX5622 after sciatic nerve injuries [44]. In this study, resident microglia were reduced by 46% after PLX5622 treatment, and the occupancy of the somatic MN surface by microglia after PNI was reduced by 60%. Whether CSF1 and CSF1-R activation are sufficient was analyzed by directly delivering CSF1 into the spinal cord. Intrathecal CSF1 induced microglia proliferation in the dorsal horn, upregulated expression of microglia activation markers, and induced mechanical hypersensitivity [57]. The ventral horn microglia reaction was not examined. Further signals have been proposed for activation of spinal microglia after PNI and close examination of published images from spinal cords with depleted CSF1 expression reveals some microglia with morphologies typical of activation, suggesting the possibility of alternative pathways [3,73,75]. The microglia transcriptome was not analyzed in any of these studies; thus, it cannot be ruled out that some microglia might undergo subtle changes in gene expression in response to injury and in a CSF1-independent manner. An alternative ligand of CSF1-R is interleukin 34 (IL-34) which was shown to exert effects on specific microglia populations, for example, white matter microglia [76]. IL-34 is constitutively expressed in sensory neurons and does not change expression after CCI [76]. It is unknown if spinal IL-34 can upregulate after other nerve injuries or if its release is enhanced without changes in gene expression.

One further possibility is the activation of receptors that converge in the same pathway downstream of CSF1-Rs. CSF1-R acts through DNAX activating protein of 12kDa (DAP12), the immunoreceptor tyrosine-based activation motifs (ITAMs), and Spleen associated Tyrosine Kinase (SYK). DAP12/ITAM/SYK activates protein kinase B (AKT), cAMP response element-binding protein (CREB), phospholipase C (PLC), Vav, and mitogen-activated protein kinase (MAPK) [77,78,79,80]. These signaling cascades influence microglia proliferation, migration, the release of cytokines, and modulate neuroinflammation and phagocytosis. DAP12/ITAM/SYK is a signaling hub shared by several surface receptors involved in microglia activation [81,82]. Triggering receptors expressed on myeloid cells-2 (TREM2), complement 3 receptors (CD11b/CD18), and some types of Fc receptors all converge in these same pathways [83]. All are upregulated in microglia after PNI. For example, TREM2 upregulation after PNI increases DAP12 phosphorylation, and blocking this mechanism reduces hyperalgesia [84]. CD11b, and more specifically the rat-specific monoclonal antibody that recognizes this surface antigen (OX-42) has been widely used to identify activated microglia in the spinal cord after a variety of nerve injuries [2]. Finally, it has also been known for some time that microglia surrounding axotomized MNs accumulate immunoglobulin G by Fc receptor binding [85]. A recent “micro-report” suggests Fc gamma receptor 1 (FcgRI) is also upregulated in the dorsal horn and participates in nerve-injury-induced chronic pain [86]. These less explored signals could contribute to microglial activation and/or modulate its properties, perhaps resulting in different behaviors and functions over different cellular elements of the dorsal or ventral horn. The ligands that activate these receptors include stress or “eat me” signals, such as phosphatidylserine for TREM2, opsonization through the classical complement pathway for C3Rs, and IgG and complement surface molecules for Fc receptors. The coordination and integration of these various signals during the diverse spinal microglia response to peripheral nerve injuries are currently unexplored. However, it is interesting that intrathecal CSF1 also regulates the expression of *tyrobp*, the gene encoding DAP12 [57], suggesting a feed-forward potentiating mechanism. In conclusion, CSF1 emerged as a master upstream regulator of spinal microglia activation pathways after PNI. These can be further modulated by convergent or alternate signals summarized in Figure 2.

### 2.3. C-C Chemokine Receptor Type 2 (CCR2) and Fractalkine Receptor (CX3CR1)

Potential roles for these receptors in microglia activation inside the spinal cord after diverse types of PNI were proposed by studies showing significant effects of agonists, antagonists, or gene knockouts and deletions principally on microglia-mediated hyperalgesia [36]. The C-C chemokine receptor 2 (CCR2) binds to several chemokines including C-C chemokine ligand 2 (CCL2) (also known as Monocyte Chemoattractant Protein 1; MCP-1) and is best known for actions recruiting CCR2-monocytes to sites of inflammation and changing capillary permeability [87]. There is no CCL2 inside the spinal cord unless upregulated in MNs and dorsal root ganglion (DRG) neurons following injury in peripheral nerves [88]. Specific expression of CCL2 by neurons axotomized after PNIs suggested it could be a possible injury-related signal that activates spinal microglia. In 2003, Abbadie et al. [89] proposed that CCL2 released by injured sensory afferents activated CCR2 on dorsal horn microglia and recruited CCR2-expressing blood macrophages into DRGs. These events were associated with PNI-induced hyperalgesia because this was diminished in CCR2 knockouts. The temporal and spatial expression of *ccl2* mRNA in sensory neurons and MNs after PNI correlated with the time course of microgliosis in dorsal and ventral horns reinforcing the idea that CCR2 might be a critical injury signal for microglia activation [90]. This was tested by analyzing dorsal horn microglia after PNI in CCR2 knockout mice and leukocyte infiltration in GFP chimeric mice [91,92]. The authors concluded that microglia CCR2 was necessary to attract peripheral blood macrophages to the spinal cord and that blocking this process prevented development of hyperalgesia.

Some of these conclusions have since been disputed. Infiltration of CCR2-expressing macrophages in DRG and peripheral nerves after PNI has been amply confirmed (the latter in response to Schwan cells’ derived CCL2 attracting CCR2-macrophages to nerve regions undergoing Wallerian degeneration), but their function at these locations is on regenerative processes, not pain [93]. In the spinal cord, we were unable to confirm CCR2 expression in dorsal or ventral horn microglia after PNI using CX3CR1-GFP CCR2-RFP double reporter mice [3]. This agrees with a previous study that focused on the dorsal horn [94]. We were also unable to find large differences in microgliosis in CCR2 knockout animals estimated by Iba1 (Ionized calcium-binding adaptor molecule 1). Iba1-expressing microglia abundance, migration towards MNs, and morphological changes were all similar with and without CCR2 [3]. Moreover, infiltration of the spinal cord by CCR2 blood-derived cells was dependent on the type of nerve injury, being significant only after full transections of large nerves (e.g., sciatic) and only in the ventral horn [3,94]. Partial reduction in CCR2-cell infiltration was observed in CCR2 knockouts but this was explained by a dramatic decrease of circulating CCR2-monocytes in the blood since CCR2 is also necessary for their egression from the bone marrow [95]. Despite having no role in microglia activation, lack of CCR2 affected some consequences of the neuroinflammatory response inside the spinal cord: it reduced the removal of proprioceptive synapses in the ventral horn and ameliorated hyperalgesia in the dorsal horn [7,91]. Interestingly, using *ccl2-RFP* reporter mice and comparing axonotmesis after sciatic nerve transection versus neurotmesis after nerve crush, we found *ccl2* expression in MNs a few days after either injury, as reported before using in situ hybridization. In addition, *ccl2* upregulated specifically in ventral horn microglia two weeks after the more severe injuries [96]. In conclusion, a CCR2-dependent mechanism modulates downstream effects of microglial activation without directly affecting their initial proliferation, migration, and morphological activation after PNI. The exact cellular elements releasing CCL2, the cell targets expressing CCR2, and the infiltration of blood-derived CCR2-cells differ between dorsal and ventral horns and depend on the type of injury, and time after injury.

Fractalkine (CX3CL1) is a unique chemokine discovered in 1997 that specifically acts on one receptor, CX3CR1 [97]. Fractalkine is expressed on the surface of neurons and its interaction with CX3CR1 maintains microglia in a surveillance state. Therefore, a logical hypothesis was that disruption of this mechanism would enhance microglia responses to any insult. However, interpretation of loss-of-function manipulations is complicated given the redundancy of signals that dampen microglia activation (CD200-CD200R; CD47-CD172a/Sirpa) [87,98]. These might obscure the impact of the removal of any one in isolation. The generation of the widely used CX3CR1-GFP knockin-knockout mouse [99] established that this receptor is expressed by microglia, perivascular and meningeal macrophages, and some circulating monocytes, and allowed experiments to test possible functions by visualizing microglia (GFP-labeled) interacting with stressed neurons in CX3CR1 knockouts (*cx3cr1^GFP/GFP^* homozygotes). The original study analyzed the microglia reaction around axotomized MNs after injuring the facial nerve reporting a normal microglial reaction in time course, proliferation, and differentiation [99]. Nevertheless, CX3CR1 expression is augmented after PNI as revealed by increased *cx3cr1* mRNA in the spinal cord dorsal horn and around axotomized MNs in the facial nucleus [100] and increased fluorescence in dorsal and ventral horn microglia in *cx3cr1*^GFP/+^ mice (Figure 3). IL-6 has been proposed to regulate CX3CR1 expression in the spinal cord after PNI [101]. Interestingly, MN IL-6 simultaneously modulates several other markers of microglia activation around axotomized facial MNs, but without being necessary for proliferation, migration towards MNs, or development of activated morphologies [102]. We observed a tendency toward larger numbers of microglia in the ventral horn after PNI in *cx3cr1^GFP/GFP^* knockout mice, but this did not reach significance (Figure 3). Despite CX3CR1 abundance in the ventral horn (more microglia and increased protein expression per cell), fractalkine (*cx3cl1*) mRNA expression is reduced in axotomized MNs [100]. It is unlikely CX3CR1 modulates the initial events of microglia activation in the ventral horn and its significance for microglia surrounding axotomized MNs needs further exploration.

CX3CR1 mechanisms differ in the dorsal horn. Induction of mechanical and thermal hyperalgesia by intrathecal injection of fractalkine and a reduction in PNI-induced hyperalgesia in CX3CR1 knockout animals have been consistently reported [94,103,104]. In contrast to axotomized MNs, uninjured dorsal horn neurons do not downregulate fractalkine mRNA after PNI [105] and enzymes that release fractalkine from the membrane are necessary for the development of hyperalgesia after PNI [103]. Fractalkine is a transmembrane protein that can exist in soluble form released from cell surfaces by enzymatic cleavage by either tumor necrosis factor-alpha (TNF-α), metalloproteinase domain-containing protein (ADAM10), or cathepsin S, all enzymes expressed by activated microglia. Cathepsin S was shown to be necessary for CX3CR1-induced hyperalgesia after PNI suggesting fractalkine actions in the dorsal horn are secondary to its release [57]. Cathepsin S expression is upregulated in spinal microglia by CSF1 [57]. Concurrently, fractalkine cleavage from the membrane frequently occurs in response to excessive excitatory input [106], and cathepsin S cleavage and release of fractalkine in the dorsal horn requires high levels of ATP acting on P2X receptors [107]. CX3CR1 activation by soluble fractalkine induces microglia chemotaxis and activates downstream pathways, importantly MAPK leading to IL-1β, TNF-α, and IL-6 release and pro-nociception [108,109]. IL-1β, TNF-α, and IL-6 together increase AMPA/NMDA currents while decreasing GABA/Glycine synaptic mechanisms on dorsal horn interneurons and this creates a hyperexcitable network [110]. The CX3CR1 mechanisms at play in the dorsal horn, contrast with the lack of discernible effects in the ventral horn. Whether this is explained by different actions of released and membrane-bound fractalkine will need to be investigated further. There is precedent in differential regulation of intracellular calcium in microglia depending on the type of CX3CR1 activation. Activation of CX3CR1 with membrane-bound fractalkine during microglia-neuron cell-cell interactions generates larger calcium responses than when evoked by soluble fractalkine in unattached microglia [100]. Thus, CX3CL1-CX3CR1 actions in dorsal versus ventral horns might differ due to different types of interaction and sources and availability of fractalkine. In any case, CX3CR1 is dispensable for activation and proliferation of dorsal and ventral microglia, although it modulates microglia function and properties, especially in the dorsal horn.

### 2.4. Toll-Like Receptors (TLRs)

Toll-Like Receptors (TLRs) are expressed in a variety of cells and respond to molecules referred to as pathogen-associated molecular patterns (PAMPs) in case of infection, or to damage-associated molecular patterns (DAMPs) in case of injury in non-infectious conditions. All known TLRs are expressed by microglia [111], but TLR2 and TLR4 are the two most studied after PNI or concerning chronic pain. One of the earliest indications of microglia involvement in the development of chronic pain was a 1994 study that induced mechanical allodynia in otherwise uninjured animals after intrathecal administration of lipopolysaccharide (LPS), a TLR4 ligand, or zymosan, a TLR2 ligand [112]. The role of TLR4 in chronic pain is now well established but differences between the sexes in mice have been frequently noted. Several studies attenuated intrathecal LPS-induced or PNI-induced allodynia by genetic or siRNA knockdown of TLR4 or pharmacological antagonism, and usually (but not always) these studies reported the results to be specific to males, not female mice [113,114,115,116]. One limitation was the interpretation of the exact location of actions after a global block of TLR4 mechanisms, given their widespread expression. To target TLR4 in specific cell types, two recent studies analyzed cell-specific TLR4 deletions in either microglia (using the CX3CR1 promoter) in a pain model of orthopedic trauma (tibial fracture) or in C-fiber nociceptors (using Nav1.8 promoter) to study mechanical allodynia after PNI. Both studies showed that TLR4 in either microglia or nociceptors influences pain but while microglia TLR4 was more significant in males, the opposite was true for C-fiber nociceptor TLR4 [117,118].

Demonstrations of TLR2 involvement in hyperalgesia after PNI followed a similar model by studying global TLR2 knockout mice, but in this case, a controversy was raised by differing results depending on the type of nerve injury. Studies that used proximal nerve transections (typically L5 spinal nerve) demonstrated an attenuation of pain in TLR2 knockouts as well as reductions in the expression of the proinflammatory pain mediators IL-1β, TNF-α, BDNF, and iNOS [119,120]. This was not the case after mid-thigh sciatic nerve ligations in a CCI model [121], suggesting either the necessity of axotomy and/or close proximity of the injury to the cell body to elicit a robust TLR2-mediated microglia response inside the spinal cord. Normal dorsal and ventral horn microglia proliferation was found in TLR2 knockouts after sciatic nerve CCI.

TLR2 and TLR4 can bind many different DAMPs and further insights into their roles stem from recent discoveries of possible endogenous ligands that activate these receptors after nerve injuries. The sialylated glycosphingolipid GT1b was recently proposed as a key ligand of TLR2s. GT1b production is increased in damaged sensory neurons after upregulation of sialyltransferase St3gal2 and thereafter GT1b undergoes anterograde transport to the dorsal horn and release [122]. GT1b is thus a critical DAMP released by the central terminals of injured sensory afferents and acts on dorsal horn microglia TLR2. A direct comparison of intrathecal injections of CSF1 and GT1b into the spinal cord showed complementary activation profiles in dorsal horn microglia [123]. Although CSF1 induced robust microglia proliferation and some upregulation of microglia-activated markers, GT1b induced significantly more robust upregulation of IL-1β, TNF-α, and NADPH oxidase 2 (Nox2), but without inducing microglia proliferation. The authors concluded that dorsal horn microglia are co-activated by CSF1-R and TLR2, each driving different properties of the microglia reaction after PNI. This mechanism seems specific to primary afferent activated microglia and not microglia around injured MNs, and it remains to be shown whether it is generalizable to different types of nerve injuries.

The search for endogenous ligands of spinal TLR4 activation recently uncovered high mobility group box-1 protein (HGMB1) [124]. HGMB1 is a nuclear chromatin-binding protein that translocates to the cytoplasm after cellular damage and can bind several TLRs. In a model of arthritis pain, *hmgb1* mRNA and extranuclear protein levels were increased in the dorsal horn, particularly in neurons. HMGB1 induced mechanical hyperalgesia by specific interaction with TLR4, but not TLR2. In parallel to other reports, TLR4 effects had a shorter time course in female mice compared to males. In another study performed only in female rats, HGMB1 expression increased in the damaged sensory neurons after tibial nerve transection and rendered them hyperexcitable [125].

In summary, TLR action on microglia seems to be specific to injury model and sex, differing on the location and time course of release of relevant ligands. The effects also seem confined to activation of dorsal horn microglia by injured afferents. In contrast, the ventral microglia response around axotomized MNs after PNI (sciatic nerve transection) seems unaffected in TLR2 or TLR4 knockouts [126]. Nevertheless, exogenous TLR4 activation significantly alters microglia around MNs, and preconditioning spinal microglia by systemic LPS exacerbates the microglia reaction around axotomized MNs after sciatic nerve crush [127]. It also protected lumbar and thoracic MNs from ischemic damage after aortic occlusion [128]. On the other hand, no effects of genetic removal of TLR2 were detected on MN survival in a model of MN-induced cell death (adult facial nerve axotomy in mice) [129]. A possible interpretation is that while pharmacological activation of TLRs modifies ventral microglia function after PNI, normally there is limited release of endogenous TLR ligands in the ventral horn after PNI to induce significant actions.

### 2.5. Activation of Microglia by Purinergic Receptors

Signaling through purinergic receptors is critical for microglia biology. ATP is released in the spinal cord during synaptic neurotransmission [130,131,132,133], but levels increase after PNI because of non-synaptic release from injured neurons or stimulated astrocytes, with exact sources not always fully identified [134,135,136]. Microglia express several ionotropic (P2XRs) and metabotropic (P2YRs) purinergic receptors. Spinal cord microglia express the ATP-responding P2X4R, P2X7R, ligand-gated cation channels, and the ADP/UDP responding metabotropic receptors P2Y6R, P2Y12R, and P2Y13R. Released ATP is efficiently converted to ADP and eventually to adenosine that can act on P1 Adenosine receptors (AR). Spinal microglia can express all subclasses of adenosine receptors (A1, A2A, A2B, and A3). However, not all P2XRs, P2YRs, and ARs are simultaneously co-expressed in microglia or respond equally to PNI. In many brain regions, surveying non-activated microglia express P2Y12R and A3R, while expression of P2X4R and A2AR is relatively low. This pattern switches when microglia become activated, pro-inflammatory, retract processes, increase phagocytic activity, or transform into ameboid-like cells [137]. In the spinal cord, P2Y12R is expressed in non-activated microglia [138], but expression further increases after PNI along with other P2YRs (P2Y6R, P2Y13R, P2Y14R) [139,140,141]. In contrast, P2X4R and P2X7R expressions are very low in non-activated microglia, and both upregulate within a week after PNI [142]. Pharmacological or genetic block of P2YRs or P2XRs does not interfere with microglia proliferation and activation after PNI, on the contrary, microglial activation is necessary for upregulating purinergic receptors, either augmenting expression from basal homeostatic levels (P2Y12) or inducing *de novo* expression (P2Y6, P2X4, P2X7). The transcription factor interferon regulatory factor 8 (IRF8) is upregulated in microglia after PNI and drives up the expression of *p2x4r* and *p2y12r*, although not *p2x7r* [143]. IRF8 is a transcription factor critical for microglia development [98], normal gene expression in microglia [144], and is part of the CSF1-R gene regulated network [145]. Thus, P2XRs, P2YRs, and ARs are regulated in PNI-activated microglia changing their responses to evolving levels of ATP, ADP, UDP, and adenosine inside the spinal cord that, in turn, differ depending on location, release mechanism, cellular origin, severity, and time after injury. Much is unknown about purinergic mechanisms on microglia around MNs, but there is extensive literature on the dorsal horn about pain mechanisms [146,147]. Similarly, purinergic actions on microglia have been abundantly studied in many brain pathologies. This information will be used here to generate hypotheses about microglia purinergic mechanisms around axotomized MNs. For more comprehensive reviews on purinergic signaling in microglia see [1,63,146,148,149,150].

The binding of ATP/ADP/UTP to these purinergic receptors is involved in microglia directional motility and even neural wrapping in many brain regions [146,148,151,152]. Therefore, purinergic signaling could modulate the reported dynamic extension and retraction of microglia processes around injured MNs [69]. One key receptor regulating directional migration and motility is the ADP-activated G_i/o_ P2Y12R [153,154]. During initial responses to injury, P2Y12R modulates process extension towards neurons by increasing integrin-b1 at the tips of microglial processes [155]. Intriguingly, Cserép et al. (2020) found that P2Y12R also mediates microglia interactions with uninjured neurons in the cortex to monitor their metabolic status [63]. They showed that microglia membrane microdomains containing P2Y12R contact neuronal membrane spots enriched with clustered voltage-gated potassium channel 2.1 (KV2.1) that act as mitochondria anchoring hubs and regulate local membrane endo- and exocytosis. Information about mitochondria and neuronal health is transferred at these microglia-neuronal junctions which increase in size with activity and paracrine non-synaptic ATP release. Remarkably, after injury (ischemia in this study) KV2.1 de-clusters and microglia then enwrap cortical neurons through a mechanism that is also P2Y12R-dependent. In summary, microglia P2Y12Rs (and maybe P2Y13Rs of similar pharmacology and actions) monitor neuronal homeostasis and their activation induces closer microglia physical interactions. Adenosine A3Rs are also expressed by surveying microglia and bind adenosine metabolized from released ATP. A3Rs co-stimulate microglia process extension in the presence of P2Y12R [156], suggesting graded spatial interactions with neuron profiles in response to ATP/ADP/adenosine levels.

It is unknown if similar P2Y12R junctions are found at points of interaction between microglia and MNs, but there are intriguing parallels between the described interactions of microglia with cortical neurons mediated by P2Y12Rs and two-photon visualization of microglia dynamics on intact and axotomized MNs. Transient contacts on the cell body surface of intact MNs transition after axotomy to frequent interactions, sometimes micro-phagocytic, and finally to a more permanent surface covering [69]. Moreover, KV2.1 is known to de-cluster in MNs after axotomy [157], a process that was also correlated with microglia wrapping in the cortex. Once the MN is surrounded by microglia processes, primary processes become stable and less dynamic but extend multiple filopodia. A recent report showed that filopodia-mediated surveillance is associated with decreased P2Y12R signaling and increased cAMP inside filopodia [158]. In addition, the UDP-activated Gq/11 P2Y6R is also upregulated around axotomized MNs [139]. P2Y6R can mobilize internal calcium stores through inositol triphosphate (IP3) receptors and increase phagocytic activity while reducing microglia process lengths and ramification [159]. In comparison to ATP/ADP signals, UDP is released by stressed or damaged neurons, suggesting a potential signal that can modify the behavior of microglia depending on the health status of the immediate MN they associate with. Finally, neuronal injury can produce excessive ATP leakage that when converted to adenosine through ectonucleotidases, such as C39, increase local adenosine concentration. Adenosine acting on A2ARs promotes process retraction and phagocytic ameboid phenotypes. A2AR expression is very low in the spinal cord, but it is upregulated with microgliosis after PNI [160]. Changing P2YR and AR activations can explain the variety of microglia interactions described over axotomized MNs. Figure 4 graphically illustrates differences in purinergic signaling receptors in relation to microglia location in the CNS (Figure 4A), microglia state (Figure 4B), and process versus filopodia extension (Figure 4B).

In the dorsal horn, P2Y12Rs are necessary for microglia wrapping of myelinated sensory afferents injured in the periphery [70]. These microglia-axon interactions were proposed to promote hyperalgesic behaviors through P2Y12Rs and maybe other P2YRs [139,140,141]. However, the principal effect of ATP activation on dorsal horn microglia and in relation to pain mechanisms is through ligand-gated P2X4 and P2X7 receptors [142]. Both are upregulated in activated microglia and promote hyperexcitability of dorsal horn pain-transmitting interneurons by releasing BDNF that, acting on tropomyosin receptor kinase B (TrkB) receptors, results in the partial removal of chloride potassium transporter isoform 2 (KCC2) from the membrane of pain transmitting spinal neurons, without a change in *kcc2* gene expression. This, in turn, leads to less efficacious inhibitory synapses, hyperexcitability, and decreased pain thresholds [161,162,163,164]. In this model, spinal microglia respond to synaptically-released ATP from dorsal interneurons, suggesting a positive feedback loop that reinforces and maintains dorsal horn network excitability in the presence of elevated nociceptive inputs [136]. Microglial P2X4 signaling is required for BDNF release and development of allodynia after PNI in males but not in females, suggesting a significant sex dimorphism [136]. In addition to P2X4R, P2X7R is also implicated in the dorsal horn microglia response to ATP leading to hyperalgesia [165,166,167]. Persistent activation of P2X7R in TLR4-primed microglia in vitro results in pannexin conjugated membrane pores that increase calcium entry and release of IL-1β and cathepsin S [107,168], which both augment microglia actions on dorsal horn nociceptive synaptic function.

Less is known about P2XR mechanisms on microglia associated with damaged MNs. Axotomized MNs also downregulate KCC2, but by difference to the dorsal horn, injured MNs suppress *kcc2* gene expression in a cell-autonomous manner in both males and females and independent of microglia, BDNF, or TrkB [73]. It results in a total removal of KCC2 from the proximal somatodendritic membrane. In addition, there are significant alterations in synaptic composition around the cell body of axotomized MNs [42]. It is now established that microglia are not central to these synaptic changes (see below), but it needs to be investigated whether they modulate neurotransmission of remaining synapses through the release of cytokines and/or neurotrophins, as shown in the dorsal horn.

## 3. Theories about the Function of Microglia around Axotomized Motoneurons

### 3.1. Neuroprotection or Neurodegeneration

Motoneuron fate after PNI (axotomy) differs according to age, MN type, injury severity, proximity to the cell body, and species. PNI in neonates (P1–P2) causes a massive loss of MNs (>90%) in all species that is rapid, occurs within days, is independent of injury severity (crush, cut, or injury location), and is similar for spinal and brainstem facial or hypoglossal MNs [169]. In contrast, MN cell death after nerve transections in adult rats is uncommon and occurs only after extended periods in the absence of axon regeneration (e.g., after a facial or hypoglossal nerve section and resection injury [170,171]). In comparison, transection injuries to the facial, hypoglossal, or various limb nerves in the adult mouse cause significant but delayed MN loss, even in the presence of active regeneration (40–60% loss after facial or hypoglossal nerve injury and 10% loss of spinal MNs after distal limb injuries) [172,173,174,175,176,177,178,179,180]. Increased MN loss is observed with decreasing proximity of the injury to the cell bodies and with increased gaps between the nerve stumps delaying axon regeneration [179,181]. An extreme case is when motor axons are cut at their exit from the CNS (nerve avulsion) in which case significant numbers of MNs are lost within two weeks after injury in adult rats and mice (70–75% MN loss after facial nerve or ventral root avulsion [182,183,184]). The mechanisms of cell death differ in each case, in parallel with their different time courses and intensities. MN death after neonatal axotomy is rapidly induced by classical apoptotic pathways including upregulation of bcl-2-like protein 4 (BAX) and fas cell surface death receptor (FAS), activation of caspase pathways, and DNA fragmentation. This death is countered by overexpression of B-cell lymphoma-2 (BCL-2) and inhibitors of apoptosis or genetic deletion of pro-apoptotic pathways [179,185,186,187,188,189,190]. In contrast, in adult mice, but not rats, MNs express Noxa after axotomy, a p53 pro-apoptotic factor independent of BAX, which tips the balance between pro-apoptotic versus pro-survival genes with slow kinetics [176,191]. This explains the delayed MN cell death in adult mice. In the adult, MNs might also undergo cell death by necrotic mechanisms instead of apoptosis [192]. Exogenous delivery to the nerve stump or inducing expression with viral vectors of effective neurotrophic factors (BDNF, neurotrophin-3, glial cell derived neurotrophic factor, and insulin-like growth factor) or cytokines (ciliary neurotrophic factor, i.e., CNTF) prevents apoptosis, sometimes by directly reversing apoptotic pathways [193,194,195,196,197,198,199,200,201,202,203,204,205]. In general, these studies support the idea that MN death is caused by the loss of trophic support. Normal neurotropism from muscles can be replaced by supporting cells in axons after muscle disconnection in adults [179]. Thus, shorter axons provide lesser support and correlate with larger MN death. Correspondingly, supplemental neurotrophic factor treatments also ameliorate MN cell death in adult mice including after root avulsions [206]. Microglia surrounding MNs are often proposed as a source of neurotrophic factors, but MN preservation by specifically microglia-derived neurotrophic factors has not yet been demonstrated [73]. Moreover, upregulation of neurotrophic factor expression by injured spinal, facial, and hypoglossal MNs and supporting cells in injured nerves, is commonly reported (reviewed in [207]), but whether microglia surrounding cell bodies of injured MNs also upregulate and release neurotrophic factors is not as clear.

A larger microglia reaction was found around MNs undergoing BAX-independent cell death [179] and axotomized MNs in adult mice display larger coverage by microglia compared to rat MNs after similar injuries [208]. This suggests that microglia coverage might correlate with cell death susceptibility. One function of phagocytic microglia is assumed to be the removal of debris from degenerating MNs. Highly phagocytic microglia are more common after injuries inducing MN death (adult mouse, root avulsion in rat and mouse) compared to injuries that do not induce cell death (adult rat distal nerve injuries). Dying adult MNs are fully covered by clustered microglial nodules, amply described in the facial nucleus [209] and more recently in the spinal cord [3]. Microglia in these clusters display elevated levels of CD68 and are associated with T-cells. It is assumed that MNs underneath these clusters are in advanced degeneration and are being partially removed by microglia, but it is unknown if microglia contributed to the degenerative process. Phagoptosis (i.e., cell death by microglia engulfment) has never been reported for MNs after PNI. Microglia surrounding axotomized adult mouse MNs that are surviving and regenerating also express phagocytic markers that polarize towards the MN surface [44,69]. Electron microscopy analyses demonstrate that microglia regions opposite to these “healthy” but axotomized MNs remove degenerated and fragmented neuropil elements. This includes synapses labeled with markers of necroptosis, a form of cytokine-induced cell death that can affect whole neurons or specific neuronal compartments such as axons and synapses [44]. In addition, microglia uptake a variety of extracellular vesicles, some of which might be derived from the MN itself. It is unclear whether these activities are associated with MN preservation. One possibility is that microglia exert a neuroprotective function sometimes overwhelmed by mechanisms inducing cell death. Alternatively, microglia could switch between neuroprotective and pro-degenerative in response to specific microglia-MN signaling. At present, it is unknown what mechanisms differentially protect specific MNs after injuries in which some regenerate while others degenerate. Resilient MNs that survive long after nerve injury without reinnervating muscle have been identified in the spinal cord, facial and hypoglossal [177,210,211] but they have not yet been associated with any specific properties of their surrounding microglia.

A variety of manipulations have been used to disrupt the microglia reaction and subsequently test the role of microglia in the neuroprotection of axotomized MNs. In one early study, microglia proliferation after hypoglossal nerve transection in the adult rat was blocked by continuous application of antimitotic drugs that resulted in a small, but statistically, significant, protection of MNs from cell death when microgliosis was diminished [212]. The authors suggested the small effect size was due to the lack of large cell death in this model, leaving little room for significant rescue. However, modification of the microglia response using nimodipine, an L-type calcium channel blocker, did not affect MN cell death in the adult rat facial nucleus after section-resection injuries which caused larger amounts of cell death [170]. These early experiments in adult rats suggested that microgliosis had a slightly detrimental or no effect on MN survival. Experiments in mice have been more revealing because of the larger amount of induced MN death in the adult and the availability of a variety of genetic models. Transforming Growth Factor beta 1 (TGF-β1) and cathepsin S knockout mice resulted in diminished microglia responses around axotomized adult facial MNs and correlated with increased MN death. TGF-β1 is required for developmental microglia maturation, homeostasis, and activation after injury in adults [213], thus, TGF-β1 knockouts displayed reduced basal microglia and astrocytic levels and reduced microglia responses after facial nerve injury [214]. Lack of cathepsin S reduced microglia enzymatic degradation of Tenascin-R in perineuronal nets preventing close enwrapping of injured MNs by microglia [215]. Both studies suggested a neuroprotective role for microglial activation around axotomized MNs and specifically highlighted the significance of physical contact with the MN surface. This conclusion agrees with a more recent study that attenuated the microglia response to hypoglossal nerve transection in adult DAP12 knockout mice [216]. In these animals, microgliosis was similar during the first-week post-injury but significantly diminished during the second week with strong reductions in microglia numbers and lower expression of *TNF-α*, *IL-1β*, *IL-6*, and, to a lesser extent, *CD68* mRNAs. These changes were associated with reduced neuroprotection evident by larger MN loss at four and six weeks after injury.

In contrast to these reports, another study analyzed MN survival in cathepsin B knockout mice after adult hypoglossal nerve transection [217] and concluded that cathepsin B is expressed by reactive microglia and promotes MN death. In the absence of cathepsin B, there was increased MN preservation and a moderate increase in microglia numbers while expression in MNs of the autophagy marker LC3-II (microtubule-associated protein 1A/1B-light chain 3 phosphatidylethanolamine conjugate) and the pro-apoptotic transcription factor Noxa, were both decreased. Microglia-derived cathepsin B could therefore be a negative regulator of microglia neuroprotective mechanisms.

Finally, minocycline has been widely used as a suppressor of microglial activation although the interpretation of these studies includes significant concerns because of the lack of specificity and exact information about the molecular targets [218]. Nonetheless, the microglia reaction was decreased and shortened using minocycline in two studies that analyzed the neuroprotection of hypoglossal MNs injured in the adult mouse [178,219]. Surprisingly, each reached opposite conclusions regarding the role of microglia on MN survival 28 days after injury. This discrepancy might be explained by the complexities of minocycline actions and the different dosages used. In summary, most studies in adult mice support the conclusion that a normal microglia reaction promotes the survival of adult axotomized MNs, but this effect is incomplete and in certain circumstances, microglia might express factors opposing this action.

### 3.2. Synaptic Plasticity around Axotomized Motoneurons

Microglial actions on “synaptic stripping” are frequently proposed to facilitate neuroprotection, but this often-cited interpretation has some important caveats. Synaptic stripping mechanisms are complex and were recently reviewed in great detail [42]. Briefly, they consist of preferential removal of excitatory synapses compared to inhibitory synapses specifically from the cell bodies of axotomized MNs. This process alters the excitatory/inhibitory (E/I) balance with some variations depending on the type of MN (i.e., hypoglossal MNs retain most inhibitory synapses [220,221]) or injury (ventral root avulsion causes a larger E/I imbalance on spinal MNs [222] than a distal tibial nerve cut [223]). Independent of its variability, synaptic stripping always inverts the E/I balance, and this action was proposed to downregulate MN activity and re-direct resources towards a regenerative metabolism [224,225,226]. Higher MN preservation following E/I inversion also seemed to agree with an excitotoxicity model for cell death of axotomized MNs [227,228,229,230,231]. However, new, and old evidence argue against these hypotheses. First, GABA/Glycine synapses on axotomized facial, hypoglossal, and adult spinal MNs become excitatory after downregulation of KCC2 [73,232,233]. Second, MN activity promotes, not decreases, regeneration in adults [234,235]. Third, neurotrophins such as BDNF and neurotrophin-3 increase MN survival (principally in neonates) but also preserve synapses on adult MNs [42]. Fourth, microglia are actively engaged in the removal of synaptic debris from the surface of MNs [44] but are not necessary for “synaptic striping” of axotomized MNs [3,44,71,236,237]. Furthermore, in the same MN axotomy model (adult hypoglossal nerve) synaptic stripping and diminished synaptic function are larger in rats than in mice, but there is increased MN survival in rats although there is lower microglia coverage in this species [208]. On the other hand, many manipulations have been performed to modify synaptic stripping intensity or the resulting E/I ratios (reviewed in [42]), and few reported any change in MN survival [238]. In contrast, larger preservation of specifically GABAergic/glycinergic synapses on spinal MNs has been consistently related to faster regeneration, muscle reinnervation, and motor function recovery [239,240,241,242].

### 3.3. Microglia and Immune System Responses around Axotomized Motoneurons

Microglia can influence MN stability by their regulation of the neuroinflammatory environment surrounding MNs which is graded to the type of injury and changes with time after injury. Because the injury is restricted to motor axons in the facial nerve, the facial nucleus of the mouse has been a favorable model to analyze time-dependent properties of neuroinflammation related specifically to MN axotomy without influence from injured sensory neurons. Two neuroinflammatory phases were distinguished. Anti-inflammatory cytokines are expressed within hours after injury and include upregulation of IL-6 in axotomized MNs and astrocytes and TGF-β1 in microglia [243]. The second phase peaks 14 days post-injury and occurs during the time of MN cell death in adults and is associated with continued expression of TGF-β1, and additional expression of the pro-inflammatory cytokines IL-1β, TNF-α, and interferon-γ [209]. All these cytokines have been investigated for their influence on MN survival. Microglia is the major source of IL-1β which requires cleavage to its biologically active form by the IL-1β-converting enzyme (ICE) [244]. ICE knockout mice show exaggerated MN cell death following facial nerve axotomy in the adult, but there was no effect in cell death after neonatal axotomy [245]. One explanation is that cell death at each age differs in mechanism and neuroinflammatory environment. However, a previous report from the same group successfully attenuated MN losses after neonatal facial nerve injury using an ICE antagonist [246]. The different results might be a consequence of different levels of ICE blockade: total and permanent in knockout animals, partial and transient using pharmacology. IL-1β is released by DAP12 activated microglia [216] and was proposed to act on IL1R1 (IL-1 Receptor type 1) upregulated in axotomized MNs [247]. However, no change in MN loss after adult facial nerve transection was detected in IL1R1 knockout mice [248]. In this same study, a dual knockout for TNF-R1&2 significantly attenuated MN death and microglia cluster/nodules disappeared from the facial nucleus suggesting that TNF-α actions increase MN death. A similar conclusion was reached after neonatal facial nerve transection in transgenic mice that overexpress a soluble form of the TNF receptor reducing TNF-α availability [249]. 

Interferon-γ is upregulated in facial adult MNs within 48 h of axotomy, but the intensity and duration of expression increase with time and injury severity (cut versus crush; distal versus root avulsion) [250]. Its expression was linked to nitric oxide synthase (NOS) and production of nitric oxide (NO) [251]. Induction of Major Histocompatibility Complex I (MHC-I) in microglia, facilitating antigen presentation to CD8+ cytotoxic T-cells was similarly correlated [244]. NOS can induce cell death and pharmacological blockade partially rescues MNs after adult facial axotomy. NO effects might be more detrimental after more proximal injuries [252]. Genetic deletion of interferon-γ receptors did not increase or decrease cell death after adult facial nerve axotomy [248], suggesting perhaps a relatively low expression of NO or CD8+ cytotoxic T-cell activity in this injury paradigm. In contrast to pro-inflammatory cytokines, TGF-β1 strongly promotes MN survival as shown by a four-fold increase in facial MN death after axotomy when it is genetically deleted [214]. Interestingly, in this experiment, TGF-β1 knockout mice were raised in a recombination-activating genes (RAG) deficient background which also impaired mechanisms of T-cell neuroprotection over facial MNs, as reviewed below.

A consistent difference between the neuroinflammatory reaction around injured facial, hypoglossal and spinal MNs of adult rats compared to mice is the infiltration of T-cells in mice, with an early slow entry during the first week that increases 100-fold and peaks 14 days postinjury [3,209,253,254]. T-cell infiltration has been associated with neuroprotection in the facial nucleus (reviewed in [255]). MN rescue by infiltrating T-cells requires antigen presentation in both the periphery and inside the CNS, the latter involving microglia MHC interactions [256]. After facial nerve transection, T-cells (both CD4+ and CD8+) are recruited by a CCR3-dependent mechanism [257] and modulated in part by IL-1β and TNF-α [258]. Axotomized facial MNs rapidly express the chemokine CCL11, which is known to be a ligand for the CCR3 receptor. At seven- and fourteen-days post-injury, MN expression decreases, while astrocytes significantly increase their expression of CCL11, exactly at the time when T-cell infiltration is maximal [259]. Infiltrating Th2 anti-inflammatory CD4+ T-cells are critical for the survival of MNs after facial nerve axotomy in the mouse [255,260].

Motoneuron death was increased in RAG2 knockout mice lacking B- and T-cells [261] and rescued by adoptive transfers of splenocytes from wild-type controls [173]. CD4+ T-cell release of the anti-inflammatory cytokine IL-10 was necessary for MN neuroprotection [262,263]. Facial MNs constitutively express the IL-10 receptor (IL-10R) and stimulation of IL-10R is known to rescue axotomized retinal ganglion cell neurons from apoptosis in a Signal Transducer and Activator of Transcription 3 (STAT-3) dependent manner [264]. Overexpressing IL-10 in astrocytes (GFAP-IL10 tg) rescued MNs from death in the facial nerve injury model [265]. These animals displayed increased numbers of microglia associated with MNs, lymphocyte trafficking, and microglia expression of MHC-II, CD16/32, (related to phagocytosis), and CD18 (a component of the complement C3 receptor) [265]. The increase in CD18 was associated with an increase in microglia adhesion to injured MNs [266]. Astrocyte overproduction of IL-6 had opposing effects by decreasing microglia attachment to MNs, reducing expression of C3 receptor, and decreasing CD4+ T cell infiltration, altogether resulting in more MN death [267]. These mechanisms may also be present in the spinal cord. After sciatic nerve transection in the adult mouse, CD3+ CD4+ T-cells enter the spinal cord ventral horn. These infiltrating cells were identified by CCR2-RFP genetic tagging, although this receptor was not necessary for entry [3]. In addition to CD4+ T-cells, we also found significant numbers of potentially cytotoxic CD8+ T-cells with many integrated into microglia clusters over degenerating MNs [69]. Similar microglia clusters/nodules in other pathologies provide the environmental factors for T-cells to exert cytotoxic actions [268]. Thus, CD4+ and CD8+ T-cell entry in the spinal cord might induce opposing actions on MN viability depending on specific local signals.

In summary, these complex interactions suggest ample redundancy, crosstalk, and balanced modulation of cytokine actions promoting MN death or survival depending on the injury (crush, cut proximal, distal, and timing), MN condition (age and type) and the degeneration mechanism involved (apoptotic versus necrotic). Combinatorial interactions in different neuroinflammatory milieus involving not only microglia but also astrocytes and the adaptative and innate immune systems provide a complex balance for neuroprotective or neurodegenerative environments. Overall, cell death of adult MNs after axotomy seems to be slow and influenced by neuroinflammation, while MN cell death after axotomy in the neonate is rapid, apoptotic, and triggered by MN dependence on muscle trophic factors in early life. In this context, microglia mechanisms might be best adapted to modulate MN survival after PNI in adults. It is worth noting that the microglia transcriptome and their capabilities are quite different in early postnatal life compared to mature microglia [268,269] and that in the dorsal horn the microglia reaction to nerve injuries dramatically differs in neonates [270].

### 3.4. Motor Axon Regeneration

MN regeneration has been well described following PNIs, however, proximity to the spinal cord and severity of the injury greatly influences regeneration success, muscle reinnervation, and functional recovery. MNs regenerate at a rate of one to three millimeters per day when conditions are favorable [271]. Intuitively, injuries that occur farther from their target muscle take longer to fully regenerate, increase the risk of failing to reinnervate the original muscle, and reduce the potential for full functional recovery [271]. Injuries that maintain intact endoneurium (axonotmesis after nerve crush) have a faster recovery and better reinnervation compared to full nerve transections [18,271]. Despite decades of research on MN regeneration following PNI, the role microglia play in MN regeneration and functional recovery remains highly debated.

The proposal that enwrapping of the cell bodies of axotomized MNs by inflammatory cells influences the regenerative process originates from a study that compared the neuroinflammatory reaction around cell bodies of sensory neurons in the DRG to MNs in the spinal cord after sciatic nerve transection [272]. This study described both cell types surrounded by neuroimmune cells of different cellular origins but similar phenotypes. The cell bodies of regenerating MNs are surrounded by MHC-II expressing microglia while sensory neuron cell bodies in the DRG are surrounded by MHC-II expressing satellite cells and peripheral monocytes. Microglia in the dorsal horn are not associated with the cell bodies of any regenerating neuron and interestingly do not express MHC-II. Based on these results and the assumption that MHC-II expression was indicative of a unique phenotype, the authors suggested that MHC-II cells surrounding the cell bodies of axotomized neurons might be involved in the regenerative process by “virtue of their capacity to release cytokines and many other mediator substances” [272]. This idea was quickly put to the test. One early study enhanced the regeneration capacity of sensory axons by inducing inflammation around DRG cells by direct infection with *Corynebacterium parvum* [273]. This effect was reminiscent of conditioning lesions in which regeneration is augmented by a nerve crush performed two weeks before and distal to the test lesion [274].

In a series of more recent papers (reviewed in [93]), it was shown that CCR2-expressing macrophages infiltrate the DRG after cut or crushed nerve lesions, surround the cell bodies of sensory neurons, and facilitate the regeneration of sensory axons in the peripheral nerve. This response was mimicked by direct injection of CCL2. In contrast, blockade of CCR2 activation reduces regeneration and prevents the effects of conditioning lesions. These studies strongly suggested that CCR2-expressing macrophages surrounding the cell bodies of axotomized sensory neurons promote regenerative capacity, but this has been more difficult to demonstrate for microglia surrounding axotomized motoneurons.

Some evidence suggests that the microglial reaction either has no effect or slows neuronal regeneration. Preventing the microglial reaction in the hypoglossal nucleus following axotomy of MNs did not affect axon regeneration and muscle reinnervation [275]. Similarly, the regenerative properties of axotomized facial MNs in *op/op* mice lacking CSF1 or mutant mice lacking the tyrosine phosphatase SHP1, an adaptor protein for CSF1-R function, were unchanged despite much-abated microglia activation in both mice [71,276]. Comparatively, mice lacking pituitary adenylyl cyclase-activating peptide (PACAP), a neuropeptide upregulated in axotomized MNs, showed an exaggerated microglial reaction with an accelerated expression of TNF-α, IL-6, and IFN-γ, and this correlated with slower regeneration [277]. The authors carefully clarified that further experiments were needed to establish a causal relationship between enhanced microglial activation and regeneration deficits [277]. However, a similar conclusion was reached in a recent study that found that axotomized MNs after a sciatic nerve cut, express and activate a neuroinflammasome (NLRP3) that is responsible for IL-1β cleavage and release after injury. Global pharmacological antagonism of neuroinflammasome resulted in a reduced microglia response centrally and faster recovery of function peripherally with more regenerating axons [278].

In contrast, other studies suggested that microglia promote regeneration. Deletion of IL-6 reduced recruitment of CD3+ T-cells and microglia activation with a modest slowing (12–14%) of axon growth in the facial nerve four days after nerve crush injury [102]. Particularly intriguing were experiments in which axons of central CNS neurons, that have little regeneration capacity after axotomy, were redirected to peripheral nerve grafts. In this condition, these central axons exhibited significant growth, which was correlated with a robust microglia reaction around the neuronal cell bodies [279]. This observation reinforced the conclusion that microglia enwrapping could be a necessary step for axon regeneration. Alternatively, it could be that neurons with axons entering peripheral nerve environments trigger retrograde microglia-activating mechanisms in the neurons, independent of the regenerative response. A more recent study suggested that CNTF released by microglia in close contact with axotomized hypoglossal MNs might promote axon regeneration while other activated microglia nearby exert opposite effects by the release of IL-1β and TNF-α [219].

In summary, the roles of central neuroinflammation and the microglia response in the regenerative process are still unclear. Most studies argue that a central microglia response is not necessary for regeneration but that some signaling that occurs during the microglia response might promote regeneration while microglia overactivation, as it occurs during prolonged denervation, might be detrimental. In this context, it is interesting to point out that the regenerative capacity of motor axons declines with time after injury, although more commonly this action has been related to the loss of regenerative promoting mechanisms derived from Schwann cells in the periphery [280,281]. More refined experiments specifically targeting central microglia around injured MNs might be necessary to resolve the exact role of microglia in regeneration.

## 4. Theories behind Microgliosis around the Central Branches of Sensory Afferents Injured in the Peripheral Nerve

Much attention has been devoted to the response of microglia in the dorsal horn in relation to pain mechanisms. These studies were mentioned earlier while discussing microglia activation mechanisms and there are many recent and thorough reviews from some of the key primary authors [9,282,283,284,285,286,287,288,289]. We will not discuss them further. A second topic amply debated in the past is that of the role of microglia on synaptic plasticity around axotomized MNs involving inputs mostly originating in uninjured spinal interneurons and descending spinal projections. This phenomenon and its functional significance were also recently reviewed at length [42]. Here, we will focus on the relation between activated microglia and the central axons and synapses of sensory afferents injured in peripheral nerves; a topic that has received comparatively less attention.

Microglia reactions in projection areas of sensory afferents injured in peripheral nerves have been analyzed after injuries to branches of the trigeminal nerve or spinal nerves. Early studies described the microglia reaction after unilateral sciatic nerve transection in all spinal cord and brainstem regions receiving projections from sensory afferents of the lower body, this included dorsal horn, ventral horn, gracile nucleus, and Clarkes’ column, while similar injuries to infraorbital nerves (a sensory branch of the trigeminal nerve) induce microglial activation in all different sensory trigeminal nuclei [2,272,290]. Microglia activation in brainstem sensory areas had similar time courses to the microglia reaction in the dorsal and ventral horns of the spinal cord [68]. Many of these regions receive projections specifically from large, myelinated mechanoreceptors from the skin (gracilis nucleus) or muscles (proprioceptors) (Clarke’s column in the thoracic spinal cord). Mechanisms of activation are likely comparable to the ones reviewed above for microglia associated with injured pain afferents or injured motoneurons, but they have not yet been specifically investigated. The function of activated microglia in these regions remains unclear. Despite extensive literature regarding microglia modulation of nociceptive transmission from unmyelinated nociceptive afferents, comparatively less is known about the actions of microglia around large sensory afferents transmitting either cutaneous mechanoreception (touch) or muscle proprioception.

After a sensory nerve axotomy, the central projections of large, myelinated axons in the dorsal horn and trigeminal nucleus frequently undergo transganglionic changes including degeneration [5,291,292,293,294]. Using electron microscopy, it has been possible to visualize microglia engulfing degenerating profiles that resemble synapses in these regions, but these phenomena were infrequent [291,293,295]. More commonly, microglia use a P2Y12 mechanism to surround myelinated axons assumed to originate from sensory afferents injured in the periphery [70]. This microglia response might be related to transganglionic degeneration of central axons or the reorganization of central somatotopic circuits in response to injury. It is well known that PNI causes significant plasticity along central somatosensory pathways to adapt to changes in innervation patterns in the periphery [226]. A recent study injured the infraorbital branch of the trigeminal nerve and demonstrated that the brainstem microglia reaction at the point of entry of injured sensory afferents in the trigeminal is necessary for synaptic plasticity and circuit reorganization along the full central pathway carrying vibrissae somatosensory information to higher centers like the thalamus [296].

Another example is the loss of Ia afferent projections to the ventral horn of the spinal cord after nerve injury. Ia afferents carry information from muscle spindles about the change in muscle lengths during motor actions and establish synaptic circuits in the ventral horn that have a very specific musculotopic organization to finely coordinate MN firing and muscle activities. Ia afferents connect monosynaptically with MNs (this is the basis of the stretch reflex), but only with those that innervate the same muscle and with a few others of similar biomechanical actions. Specificity in these circuits is established during early development by complex molecular interactions and guidance systems that do not exist in adults [297]. Remarkably Ia afferent synapses on MNs are permanently lost after transection nerve injuries, but not after axotomy due to nerve crush [37,223,298,299]. Lost Ia afferent synapses are never recovered, even after successful regeneration in the periphery. One difference between both types of injury is the poor specificity in muscle reinnervation obtained after nerve transection, for both motor and proprioceptive sensory axons [300,301]. Therefore, we proposed [42] that one possible explanation for the loss of these central connections is compensation for the lack of specificity after muscle innervation, *de facto* scrambling the organization of Ia-MN connections inside the spinal cord. This would render these synapses unnecessary or even dysfunctional after peripheral nerve regeneration. However, Ia afferent synaptic collaterals (from the same axons) to Clarkes’ column, the origin of ascending proprioceptive pathways for hindlimbs, are less affected. This may be because ascending projections do not require the specificity of connections found in the ventral horn. For example, spinocerebellar projecting neurons receive convergent proprioceptive input from many muscles and different kinds of receptors [302].

Interestingly, activation of ventral horn microglia and a CCR2 mechanism are both implicated in the permanent loss of Ia synapses and die-back of Ia axon collaterals in the ventral horn [3]. This is a striking example of local microglia activation and neuroinflammation grading according to injury severity predictive of regeneration specificity (proximal versus distal, cut versus crush) resulting in deletion of specific central branches from single afferents injured in the periphery depending on their location and function. Thus, two possible functions emerge for microglia activation in the projecting areas of large cutaneous mechanoreceptors and muscle proprioceptive afferents injured in peripheral nerves. First, some microglia might recognize and remove sensory myelinated axons and synapses that are degenerating or incapable of regenerating and reinnervating peripheral targets. Second, microglia might facilitate pseudo-adaptative synaptic plasticity of central circuits that while resulting in some loss of normal functionality, compensate for suboptimal efficiency and specificity of peripheral axon regeneration and target re-innervation.

### Complement as a Targeting Mechanism for Synaptic Pruning

Microglia synaptic pruning during development and following neuropathology is dependent on classical complement pathways requiring C1q (Complement component 1q) [303,304]. Complement mechanisms involve a set of proteins that, when expressed on synaptic terminals, mark them for recognition and degradation through microglial phagocytosis [305,306,307,308,309,310]. This process is known as opsonization and allows specific synapses to be tagged for removal by effector cells, typically a macrophage such as microglia or blood-derived leukocytes. The “classical complement cascade” is initiated by the accumulation of C1q protein on the surface of the targeted synapse and C3 and C5 recruitment. Once linked through a series of enzymatic and cleavage processes, C3 and C5 proteins promote synaptic elimination.

Complement proteins are expressed around MNs following PNI. An early study in adult rats showed a significant increase in perineuronal expression of C3 in the first two weeks after complete transection of the hypoglossal nerve [311]. In parallel, spinal and brainstem microglia activated after PNI strongly upregulate C3 receptors (CD11b, OX-42). Following sciatic nerve transection, the complement proteins C1q and C3 are found in the gracile nucleus, the dorsal horn [312], and the ventral horn [239]. C1q is expressed and released by microglia, while C3 is preferentially expressed by axotomized MNs and sensory afferents.

To investigate the role of complement in synaptic removal, Berg et al. [239] performed sciatic nerve transections in mice lacking either C1q or C3. They found after transection a 32.3% reduction in synaptophysin immunoreactivity around axotomized MN cell bodies in wild-type mice, but only a 6.2% reduction in C3 knockout mice. The main synaptic inputs rescued in C3 knockout mice were inhibitory synapses defined by expression of the vesicular inhibitory amino acid transporter (VIAAT) with a trend, that did not reach statistical significance, toward the preservation of glutamatergic synapses derived from excitatory interneurons and descending inputs (labeled by the vesicular glutamate transporter isoform 2; VGLUT2) [239]. There was no preservation of VIAAT or VGLUT2 synapses in C1q knockout mice, suggesting C3 action on inhibitory synapses is mediated through the C1q-independent alternative complement pathway. Although this study shows evidence of complement involvement or lack thereof, in the removal from the MN cell body of synapses originating in uninjured presynaptic neurons, it did not address whether the complement system is involved in the deletion of synapses from injured sensory afferents, like Ia afferent synapses. These synapses express the isoform 1 of the vesicular glutamate transporter (VGLUT1) [313]. Interestingly, C1q is necessary for the removal of Ia afferent VGLUT1 synapses from MNs in neonates during developmental synaptic pruning and proprioceptive circuit refinement in the ventral horn [314]. The potential for VGLUT1 synapse removal in adults by complement tagging is also supported by a recent report that investigated VGLUT1 synaptic removal in the hippocampus after exposure to West Nile Virus [315]. Whether complement mechanisms are involved in the plasticity of synaptic arbors from mechanoreceptors and proprioceptors after PNI in the adult remains to be investigated.

## 5. Conclusions

PNI-induced microglia reactions in the spinal cord and brainstem are variable, complex, and dynamic. There is a large variety of activation mechanisms that interact with each other and result in different microglia phenotypes that dynamically change with time after injury and injury properties. These induce a diversity of temporally regulated microglia actions in MN preservation and regeneration or trigger permanent circuit changes that can render spinal networks hyperexcitable, like in the dorsal horn (and also trigeminal [316]), or might cause circuit plasticity in the ventral horn that is pseudo-adaptative to new patterns of innervation in the periphery after regeneration.

## Figures and Tables

**Figure 1 cells-11-02083-f001:**
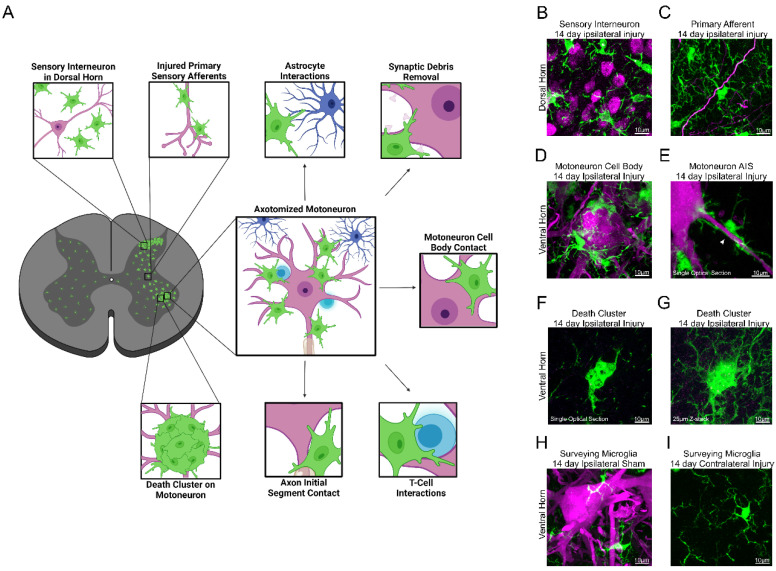
Diversity of microglia interactions with neuronal and immune cells within the spinal cord after peripheral nerve injuries. (**A**). Localization of activated microglia in the spinal cord after peripheral nerve injury and interactions with different cellular and subcellular elements. Microglia in green, neurons in magenta, astrocytes in periwinkle, T-cells in blue. (**B**). Confocal image of wild-type C57BL/6J mouse with IBA1 antibody microglia labeling (green) and NeuN antibody sensory neuron labeling (magenta)(2D projection of 26 µm thick z-stack). Image created in Biorender. (**C**–**E**). Confocal images of *Tmem119^eGFP^* genetically labeled microglia surrounding motoneurons and sensory afferents axotomized in the nerve, which were previously transduced with AAV1-tdTomato. (**C**). Microglial interactions with primary afferents in the dorsal horn (2D projection of 50 µm thick z-stack). (**D**). Microglia surrounding a motoneuron cell body (2D projection of 50 µm thick z-stack). (**E**) Microglial wrapping the axon initial segment (AIS) of a motoneuron (single optical plane, arrow indicates AIS). (**F**,**G**). Confocal images of *Tmem119^eGFP^* genetically labeled microglia forming a death cluster around a degenerating motoneuron previously labeled with Fast Blue (**shown in magenta**) (**F**) single optical plane, (**G**) 2D projection of 25 µm thick z-stack). (**H**,**I**). Confocal images of *Tmem119^eGFP^* genetically labeled microglia with AAV1 tdTomato neuronal labeling. (**H**). Surveying microglia (green) near uninjured motoneurons (magenta) (2D projection of 50 µm thick z-stack). (**I**) Surveying microglia (green) in the contralateral ventral horn of an axotomized animal (2D projection of 50 µm thick z-stack).

**Figure 2 cells-11-02083-f002:**
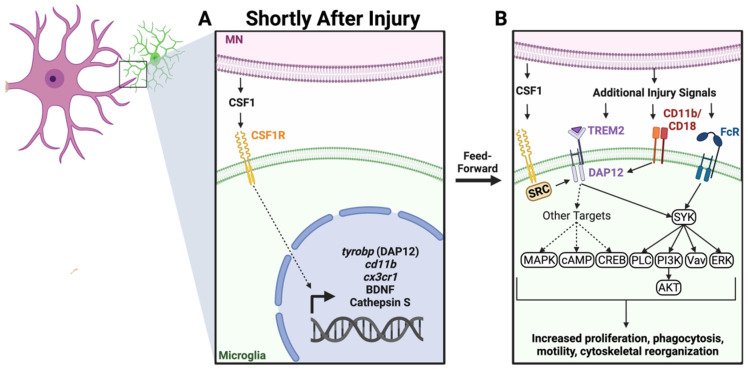
Colony Stimulating Factor 1 (CSF1) is a central signal for activation of spinal microglia following peripheral nerve injury. (**A**). CSF1 Receptor (CSF1R) activation is upstream of most signaling cascades which govern activated microglial morphology and function. (**B**). Several other receptors converge with CSF1R in similar intracellular signaling pathways. Image created in Biorender. Abbreviation key: Brain Derived Neurotrophic Factor (BDNF), Triggering Receptor Expressed on Myeloid cells 2 (TREM2), Fc Receptor (FcR), DNAX activating protein of 12 kDa (DAP12), Spleen-associated Tyrosine Kinase (SYK), Mitogen Activated Protein Kinase (MAPK), cyclic AMP (CAMP), cAMP Response Element Binding protein (CREB), Phospholipase C (PLC), Phosphoinositide 3 Kinase (PI3K), Extracellular Signal-Regulating Kinase (ERK), protein kinase B (AKT).

**Figure 3 cells-11-02083-f003:**
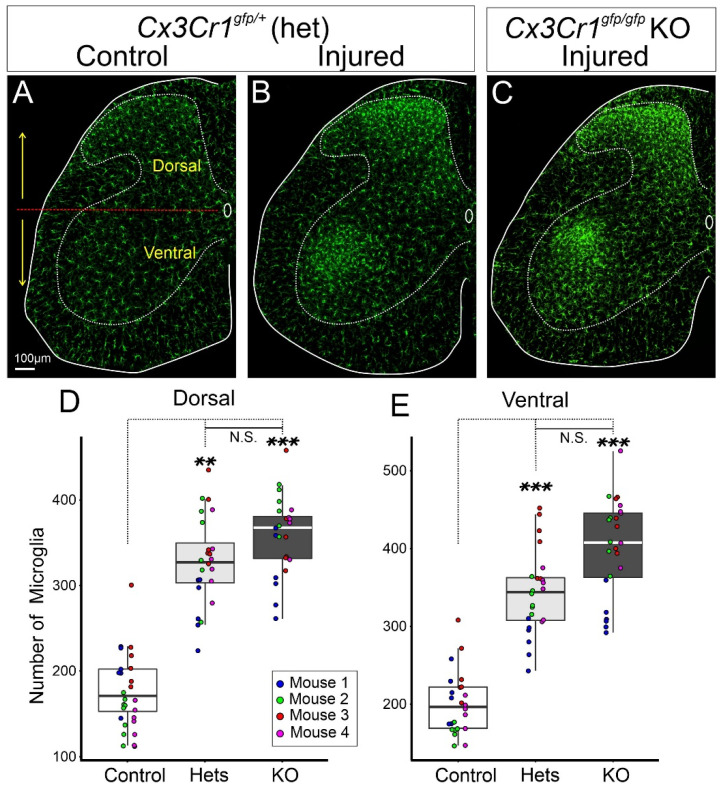
Lack of effect of CX3CR1 genetic removal in spinal cord microgliosis after sciatic nerve injury. (**A**–**C**). Low magnification confocal images showing CX3CR1-GFP microglia in a naïve spinal cord (**A**), 14 days after a sciatic nerve injury (**B**) or after the same injury in a homozygote CX3CR1 knockout mouse (**C**). The boundaries used to delimit dorsal and ventral horns are indicated by a red dash line drawn from the top of the central canal. (**D**,**E**). Quantification of microglia in dorsal and ventral horns of 50 µm thick sections of Lumbar 5. Four animals were analyzed for each phenotype. The number of microglia was automatically counted using the spot function in Imaris in five to seven sections per animal. Boxplots indicate the median (thick line), the box represents the 25th (Q1) and 75th (Q3) quartiles, and the whiskers represent the min (−1.5* Q1) and max (1.5* Q3). Each dot is a section color-coded for each animal. Following sciatic injury there is a significant increase in microglia in both dorsal and ventral horns of heterozygotes (Hets) and knockout (KO) mice. Pair-wise comparisons showed that Hets and KOs were significantly different to controls in both dorsal and ventral horns, but there was no statistical difference between them. For statistics, we used linear mixed-effect models (lme4 and emmeans packages) in R to account for variability between animals (shown by color segregation in each genotype). ** *p* < 0.001, *** *p* < 0.0001.

**Figure 4 cells-11-02083-f004:**
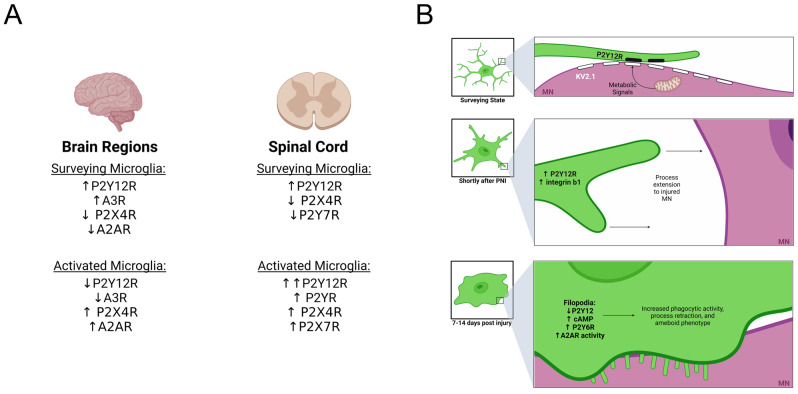
Purinergic signaling in spinal cord microglia following peripheral nerve injury. (**A**). Differences in purinergic receptor expression between microglia located in the brain versus spinal cord. (**B**). Possible purinergic signaling mechanisms between microglia (green) and motoneurons (magenta) during a surveying state, shortly following peripheral nerve injury, and a few days post-injury. Abbreviation key: ionotropic purinergic receptors (P2XRs), metabotropic purinergic receptors (P2YRs), adenosine receptors (ARs), voltage-gated potassium channel 2.1 (KV2.1), motoneuron (MN), peripheral nerve injury (PNI), cyclic adenosine monophosphate (cAMP). Images created in Biorender.

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
