# Peer review of "The Role of Microglia in Neuroinflammation of the Spinal Cord after Peripheral Nerve Injury"

_cells, 2022, doi:10.3390/cells11132083_

Round 1
Reviewer 1 Report
This is an excellent and timely review on the role of microglia in the context of the neuroinflammatory reaction occurring in the spinal cord after peripheral nerve injury. Authors provide a comprehensive analysis of mechanisms leading to microglial activation and its function in the processes of neuroprotection/neurodegeneration, synaptic plasticity around injured motoneurons, and the reorganization of spinal circuits following peripheral nerve injury. The review summarizes past controversies and analyzes recent data regarding the diversity and significance of microglial-motoneuronal interactions after peripheral nerve injuries. Overall, the review is very well written and illustrated, and sheds light on the complex phenomenon of spinal microgliosis resulting from peripheral nerve injury.
Author Response
We thanks reviewer 1 for his/her support. We have reviewed the whole manuscript for grammar spell check. We hope we have not missed any errors
Reviewer 2 Report
Comments and Suggestions for Authors
The review by Pottorf, et al., attempts to provide a comparative analysis of microglia activation in the dorsal and ventral horns in response to peripheral nerve injuries. The authors do an excellent job in providing a thorough and comprehensive review of the in vivo and in vitro studies pertaining to this topic.
The authors do an excellent job in presenting important studies in the field. However, the review becomes difficult to follow in some areas due to the density of information. Some of the sentences seem to be run-on and make the reading a little difficult to follow. For example, the second paragraph in section 3.4 (lines 906 -968) is extremely long. The three viewpoints being discussed are difficult to distinguish. Breaking the paragraph up would very helpful and make the reading more enjoyable.
Overall, it is a very comprehensive and informative review.
Author Response
We thanks Reviewer 2 for his/her support and we are very please the manuscript was found very useful. Upon re-reading we agree that we did not do our best with section 3.4. We made the following changes:
- Section 3.4, lines 928-977 have been reorganized and edited for improved clarity. We found the three different viewpoint relatively redundant, and obvious. Therefore we deleted the initial summary sentences for clarity and to move more directly into the data supporting them.
- The whole manuscript has been revised to improve reading by shortening sentences and opening new paragraphs when necessary or possible.
Reviewer 3 Report
Minor comments:
The Introduction need not have sub-parts stating introduction to different sites of injury- This can be edited with only the sites of injury as the sub-heading
Fig. 1 : panel B-I has lost clarity, specifically the texts. Figure legends need to be bold
Fig.2 can be depicted as A and B
Fig. 4: Check the text for the figure panels A and B
Author Response
We thank Reviewer 3 for his/her suggestions. We have modified the manuscript according to all and we believe this improved readibility and clarified some issues in the figures.
- Section 1 (Introduction), subheadings were revised to remove the term “Introduction to” (Lines 46, 106, 132).
- Figure caption 1, the title of the figure, and lettering have been changed to bold for consistency (Lines 243-261).
- Figure 2 has been updated to be panels A and B. Publication rights for the new figure were obtained. The figure legend has been updated appropriately (Lines 337-347).
- Figure 4A and 4B were referenced specifically in the text (Lines 619-620).